# Effect of Dietary Patterns on Inflammatory Bowel Disease: A Machine Learning Bibliometric and Visualization Analysis

**DOI:** 10.3390/nu15153442

**Published:** 2023-08-03

**Authors:** Haodong He, Chuan Liu, Meilin Chen, Xingzhou Guo, Xiangyun Li, Zixuan Xiang, Fei Liao, Weiguo Dong

**Affiliations:** 1Department of Gastroenterology, Renmin Hospital of Wuhan University, Wuhan 430060, China; 2018305230050@whu.edu.cn (H.H.); 2021103020024@whu.edu.cn (C.L.); 2016302180077@whu.edu.cn (M.C.); 2018305230044@whu.edu.cn (X.G.); 2017302180028@whu.edu.cn (X.L.); 2017302180046@whu.edu.cn (Z.X.); 2Wuhan University Shenzhen Research Institute, Wuhan 430060, China

**Keywords:** dietary patterns, inflammatory bowel disease, bibliometric, CiteSpace

## Abstract

Aims: This study aimed to analyze the related research on the influence of dietary patterns on IBD carried out over the past 30 years to obtain the context of the research field and to provide a scientific basis and guidance for the prevention and treatment of IBD. Methods: The literature on the effects of dietary patterns on inflammatory bowel disease published over the past three decades was retrieved from the Web of Science Core Collection (WoSCC) database. CiteSpace, VOSviewer, the R software (version 4.3.0) bibliometrix package, the OALM platform, and other tools were used for the analyses. Results: The growth of scientific papers related to this topic can be divided into two stages: before and after 2006. Overall, the growth of the relevant literature was in line with Price’s literature growth curve. Subrata Ghosh and Antonio Gasbarrini are the authors with the highest academic influence in the field, and Lee D.’s research results are widely recognized by researchers in this field. Among the 72 countries involved in the study, the United States contributed the most, while China developed rapidly with regard to research being carried out in this area. From a regional perspective, countries and institutions in North America, Europe, and East Asia have made the most significant contributions to this field and have the closest cooperation. Among the 1074 articles included in the study, the most influential ones tended to consider the mechanism of the effect of dietary patterns on IBD from the perspective of the microbiome. Multiple tools were used for keyword analysis and mutual verification. The results showed that NF-κB, the Mediterranean diet, fatty acids, fecal microbiota, etc., are the focus and trends of current research. Conclusions: A Mediterranean-like dietary pattern may be a good dietary habit for IBD patients. Carbohydrates, fatty acids, and inulin-type fructans are closely related to IBD. Fatty acid, gut microbiota, NF-κB, oxidative stress, and endoplasmic reticulum stress are the hot topics in the study of the effects of dietary patterns on IBD and will be emerging research trends.

## 1. Introduction

Inflammatory bowel disease (IBD) is a group of diseases characterized by chronic intestinal inflammation, including ulcerative colitis (UC) and Crohn’s disease (CD), and its incidence is increasing worldwide [1]. The etiology of IBD is not completely clear, but a number of factors have been confirmed to be related to its occurrence and development, among which dietary patterns have received extensive attention as an important environmental factor [2]. Dietary patterns refer to individuals’ eating and intake patterns over a specific period of time. In recent years, studies have shown that there is a close relationship between dietary patterns and the risk and deterioration of IBD. Unhealthy dietary patterns such as high sugar, high fat, high salt, and low fiber can increase the risk of IBD [3,4]. Therefore, diet adjustment and nutritional intervention play an important role in IBD patients, which is of great significance in improving the quality of life of IBD patients, reducing disease symptoms, and preventing disease deterioration.

Bibliometrics is a quantitative analysis method that can be used to examine literature and allows us to obtain intelligent summary statistics and analyses of citation relationships, authors, keywords, and other information in the literature. In addition, bibliometric analysis can reveal research hotspots, trends, and correlations in a particular field, and it has been widely used in various disciplines, including medicine [5,6]. The role of dietary patterns in IBD is a complex and important area of research, but no bibliometric analysis has been carried out on this research topic. This paper presents a bibliometric analysis of nearly 30 years of relevant global studies to systematically understand the relationship between dietary patterns and IBD and to reveal the frontiers and directions of the research field. The aim of our study was to provide scientific evidence and guidance for the prevention and treatment of IBD. Through an in-depth analysis of the role of dietary patterns in IBD, we will be able to provide patients with personalized dietary advice tailored to their individual characteristics and disease state. In addition, our findings will provide an important reference for the development of public health policies and dietary guidelines to promote overall population gut health and the preventive control of IBD.

## 2. Materials and Methods

### 2.1. Data Source

All data in this article were retrieved on 18 May 2023 from the Web of Science Core Collection (WoSCC) database, with the following search pattern: (((((((TS=(Diet)) OR TS=(Dietary)) OR TS=(Food)) OR TS=(Feeding-Related)) OR TS=(Eating)) OR TS=(Feeding))) AND ((((((TS=(Patterns)) OR TS=(Pattern)) OR TS=(Patterns)) OR TS=(Behaviors)) OR TS=(Behavior)) OR TS=(Pattern)) AND (((TS=(Inflammatory Bowel Disease)) OR TS=(ulcerative colitis)) OR TS=(Crohn’s disease)). In addition, two paper types, articles and reviews, were selected. A total of 1092 relevant papers were obtained from this search.

### 2.2. Data Processing

For the above search results, plain text files and tab-delimited files of the full records and cited references were exported. The plain text files were analyzed by CiteSpace (version 6.1.R6) and the R language (version 4.3.0) bibliometrix package (version 4.1.2), and the tab-delimited files were used for analysis in VOSviewer (version 1.6.19) and the Online Analysis Platform for Bibliometrics (OALM) (http://bibliometric.com/, accessed on 19 May 2023). Transformation and de-duplication in CiteSpace yielded 1092 document records, of which 1074 documents from the last 30 years (1993–2023) were selected for analysis.

### 2.3. Data Analysis

The above four bibliometric analysis tools were used to study and verify the results: CiteSpace was used for the co-occurrence analysis, cluster analysis, and emergent analysis; VOSviewer for the co-occurrence analysis and cluster analysis; the bibliometrix package for frequency analysis, relational network analysis, and geographic visualization analysis; and the OALM platform for the relationship network analysis. In addition, the journal name, impact factor (IF), and journal ranking (Q1–Q4: quartile in category) were also recorded using the 2021 edition of the *Journal Citation Reports* (*JCR*). Excel was used to draw column charts, line charts, and stacked area charts. Since the countries, institutions, journals, and authors have different distributions in different fields and CiteSpace only reviews the relevant information of the literature in the past 5 years, the analysis results of the four tools will be slightly different. At the same time, we found that the R software bibliometrix package can test the data quality, so the results of the bibliometrix package will mostly prevail when there were differences between multiple tools. The results showed that the data quality was acceptable, and most of the data were of good quality (Appendix A); a follow-up analysis was then carried out.

## 3. Result

### 3.1. The Number of Publications and General Characteristics

The annual numbers of published articles and reviews obtained from WOS and CiteSpace were visualized. As of the retrieval date, there were 1092 publications, while a total of 1074 papers were published in the last 30 years. Among them, articles account for the majority. There were 742 articles, accounting for 69%, and 332 reviews, accounting for 31% of the papers (Figure 1C). The annual and total number of related studies have increased significantly in the past 30 years (Figure 1A,B). The significant decline in the volume of documents in 2023 may be due to the fact that the data only included up to 18 May 2023 (i.e., the data for 2023 are incomplete).

According to the annual number of publications, we divided the time period into two stages: the first stage is from 1993 to 2006, where the number of related publications did not increase significantly and is in the stage of slow development, and the second stage is after 2006, where the number of related publications increased significantly (Figure 1A,B). This change may be related to events such as the WHO General Assembly releasing the “WHO Global Strategy on Diet, Physical Activity and Health” in 2004 [7], and the United States Department of Agriculture (USDA) and the United States Department of Health and Human Services (HHS) releasing 2005 Dietary Guidelines for Americans in 2005 [8].

The Price literature growth curve reveals the law of scientific literature growth, which showed that the total number of papers changed with time as F(x) = ae^bx^. It can be seen that the total and the annual number of papers should increase exponentially with time. The logical regression model was constructed after excluding the publication data for 2023. The regression function of the total number of publications was y = 2E-102e^0.1192x^, and the determination coefficient R² was 0.9987 (Figure 1D). The average annual growth rate was about 14.14%. The regression equation of the annual number of literature publications was y = 7E-90e^0.1038x^. The determination coefficient R² was 0.945 (Figure 1D), and the average annual growth rate was about 11.25%. These results are consistent with the Price literature growth curve.

### 3.2. Author Analysis

According to the bibliometrix package analysis, a total of 6039 authors contributed in related research in the past 30 years, among which Professor Ghosh S. ranked first with 10 publications (0.93%) and was the most productive author. The other four of the top five authors from the literature output were Ananthakrishnan A. N., Chan A. T., Gasbarrini A., and Haller D., all tying for second place with eight articles (0.74%) (Figure 2A,B, Appendix A). The ranking of authors’ influence under indexes such as the h-index, g-index, m-index, etc., is shown in Table 1. In a comprehensive comparison, Ghosh S. and Gasbarrini A. had the most significant influence in this field, among which Professor Ghosh S. has published frequently in this field for more than 10 years (h-index: 7, g-index: 10, m-index: 0.467, total citations: 675, start year: 2009), while Gasbarrini A. has been an excellent researcher in this field in recent years (h-index: 7, g-index: 8, m-index: 1.167, total citations: 1546, start year: 2018) (Figure 2B).

To some extent, the number of citations of the author can reflect the effect of their research results on the field. Among the 1074 references included in the study, Professor Lee D. was the most frequently cited, with a total of 130 citations. In addition, Professor Lewis J. D., Professor Boutron-Ruault M. C., Professor Carbonnel F. and Professor Clavel-Chapelon F. (Figure 2D) ranked in the top five, indicating that they had a high influence on the literature on the impact of eating patterns on IBD.

According to the generalized Lotka’s law, the number of authors y who published x papers is inversely proportional to the number c of only one paper; that is, it satisfies y = cx^n^, where c and n are constants [9]. Based on the modeling and analysis of the number of authors, the regression function was y = 5661.9x^−3.635^, and the determination coefficient R² = 0.9998 (Figure 2E); therefore, the number of authors of publications related to this topic conforms to the generalized Lotka’s law.

The co-occurrence visualization of the authors shows that the authors who published a high number of articles tend to cooperate closely with other authors, and there is often a cooperative relationship among high-yield authors. In addition, authors with cooperative relationships have formed a large number of significant cooperative groups, which may be related to the fact that these authors belong to the same organization (Figure 2A,C); however, there is no obvious cooperative relationship among most research groups. The cluster analysis of the authors shows that the researchers have significantly different sub-themes, such as “human gut microbiome” and “altered gut microbiome” related to pathogenesis, and “dietary lipid fuel” and “inulin-type fructan” related to dietary content, which indicates that the relationship between eating patterns and IBD may be related to intestinal microorganisms and lipid metabolism (Appendix A).

### 3.3. Analysis of Cooperation between Countries and Institutions

The results of the bibliometrix packet analysis based on the WOSCC data show that there were papers from 72 countries in the last 30 years. Among the 1074 publications, the United States participated in 818 articles (76.16%) and is far ahead of the next countries: China (338), Italy (313), Canada (264), and the UK (242). All of these countries produced more than 200 publications, but the United States had always been in a significant leading position. On the other hand, the number of Chinese research participants has grown rapidly in the past decade with strong momentum (Figure 3B). However, the number of corresponding authors in the various countries did not correspond with the above-mentioned numbers. Among them, 224 corresponding authors (20.96%) were from the United States. This was followed by 96 from Italy (8.94%), 92 from China (8.57%), 72 from the UK (6.70%), and 63 from Canada (5.87%), all of which were more than 60 (Figure 3C). We believe that the main reason for this difference is the extensive international collaboration in the 1074 research papers, with researchers other than the corresponding authors coming from multiple countries (Figure 3C). The number of citations of each country represents the influence of its research results to a certain extent, and the country with the most citations was still the United States, which was far ahead of the other countries with 17,315. This was followed by Canada, the UK, Italy, Australia, and China. The country with the highest average number of citations was Australia, which indicates that the average influence of its research results was higher (Appendix A).

In addition, the color depth in the geographic visualization map represents the number of studies involved (Figure 3E), and the line thickness between countries represents the cooperative relationships (Figure 3E,F). The results showed that there was a cooperative relationship among all countries. Cooperation between high-output countries tended to be higher. Generally speaking, the core areas hosting the related research were North America, Europe, and East Asia. The regions with the closest cooperation were North America and Europe, and North America and East Asia.

A total of 1706 institutions in 72 countries were identified from the bibliometrix package analysis, with the University of Calgary participating in the most published articles, with a total of 44 articles (Figure 3G), followed by the University of Toronto (27), Harvard University (26), the University of Alberta (24), and the Icahn School of Medicine at Mt. Sinai (23). Harvard University and the University of Alberta were in the leading position for a long time, while the University of Calgary and University of Toronto have developed rapidly in recent years; both their published volumes surpassed Harvard University in 2021 and 2022, respectively.

According to the institutional influence analysis based on OALM, the institution with the largest number of citations is Harvard University, while the institution with the highest average number of citations was the University of Paris (Appendix A). The bibliometrix package inter-agency cooperation analysis showed that most of the relevant research institutions were universities (Figure 3H), of which Harvard University was the most central, and its cooperation with Massachusetts General Hospital was the closest, with Massachusetts General Hospital as the medical teaching and research center of Harvard University. In addition, there were also significant cooperative relationships between the University of Calgary, University of Toronto, and University of Alberta, which are all located in Canada. This shows that there is a close relationship between inter-agency linkages and geographical distance.

### 3.4. Journals Analysis

The results of the analysis based on the bibliometrix package showed that 1074 papers were distributed in 496 journals, and the largest number of relevant publications was in *Nutrients* (70, 6.52%), followed by *Inflammatory Bowel Diseases* (26, 2.42%), the *Journal of Crohn’s & Colitis* (23, 2.14%), *PLoS ONE* (23), and the *World Journal of Gastroenterology* (23), all with more than 20 articles (Figure 4A,B and Table 2). It can be seen that the number of relevant studies in the journals all started to increase significantly after 2006 (Figure 4A), which corresponds to the two stages of development of the number of publications mentioned above. According to the h-index, *Nutrients* had the highest influence in the field of this study among the 496 journals (h-index = 22). In addition, the top five h-indexes were for the *World Journal of Gastroenterology* (16), *Inflammatory Bowel Diseases* (15), the *Journal of Crohn’s & Colitis* (15), and *Gastroenterology* (14) (Appendix A).

Bradford’s law is one of the basic laws of bibliometrics. It states that scientific journals in a certain field can be divided into core areas and subsequent partitions according to the number of publications, and the number of journals satisfies 1:n:n2. Based on this, we analyzed 21 core journals (Figure 4C). Among the journals in the core area of Bradford’s law and the top five journals published above, more than half of the journals were in JCR Q1, indicating that the literature included in this analysis is of high quality (Table 2), among which the highest impact factors were for *Gastroenterology* (IF: 33.883, Q1) and *Gut* (IF: 31.793, Q1).

VOSviewer was used to construct the journal co-citation network, and the bibliometrix package was used to analyze the cited journals (Figure 4D,E). Among all the references, the number of papers from the journal *Gastroenterology* was the most, with 3351, and the number of citations was more than 1500. *Gut*, *Inflammatory Bowel Diseases*, *Nature*, and the *American Journal of Gastroenterology* were also all JCR Q1 journals (Appendix A). This shows that the 1074 papers tended to be published in high-quality and high-impact journals.

In order to determine the distribution of citing journals and cited journals, CiteSpace was used for dual-map overlay analysis (Figure 4G). The left side of the figure shows the main types of citing journals, and the right side shows the main types of cited journals. It can be seen that the citing papers were mainly concentrated in “molecular, biology, immunology” and “medicine, medical, clinical” journals, while the cited literature was mainly published in “molecular, biology, genetics” and “health, nursing medicine” journals, which belong to the fields of clinical medicine and basic medicine and is consistent with the clustering timeline analysis results of the cited journals (Figure 4F). In addition, there were a certain number of references belonging to “environmental, toxicology, nutrition”, “psychology, education, social”, and “chemistry, materials, physics” journals, indicating that environmental, social, psychological, chemical, and other factors are also involved in the influence of dietary habits on IBD.

### 3.5. Literature and Citation Analysis

In order to determine the influence of the 1074 papers included in the study, the bibliometrix package and VOSviewer were used to analyze the citations of the most locally cited documents and the most globally cited papers in these 1074 papers, and the OALM platform was used for verification. Among the 1074 papers included, “David L. A., 2014, Nature” [10] had the highest number of cross-references, up to 84 times (Figure 5A–C). Globally, “David L. A., 2014, Nature” [10] also had the highest number of citations, with a total of 5490 citations (Figure 5D). “David L. A., 2014, Nature” [10] ranked first in both rankings, indicating that this article had a great influence on both the global scientific field and this field. This mainly demonstrated that the intestinal microbiota changes with dietary habits, and animal-based-diet-induced intestinal microbiota changes may contribute to the development of IBD [10], suggesting that the intestinal microbiota is an important research direction for studying the effects of dietary habits on IBD.

The references represent the theoretical basis of related research, from which we can see the context of the research field. We used the bibliometrix package for co-citation analysis and validated the results in VOSviewer. A total of 15 articles (Figure 5E,F and Appendix A) with a total number of over 50 co-citations were included. These 15 articles can be artificially divided into three categories: (1) studies that describe the link between dietary patterns and gut microbiota. These papers showed that different dietary compositions can lead to the formation of different gut microbiota [11,12]. Changes in gut microbiota caused by animal-based diets, dietary emulsifiers, dietary fat, and other dietary patterns may contribute to the development of IBD [10,13,14]. (2) Studies that mainly discuss a variety of dietary patterns related to the risk of IBD. These studies demonstrated that high intake of protein, sugar, polyunsaturated fatty acids, ω-6 fatty acids, and alcohol increases the risk of IBD, while intake of high-fiber foods such as vegetables and fruits is associated with a reduced risk of IBD [15,16,17,18,19]. (3) There is also a smaller category with papers that emphasize the epidemiological status of IBD worldwide and the need for innovation in prevention and treatment [1,20].

Cluster analysis was performed on the references, and their node centrality and burstiness were calculated (Figure 5G). The red nodes represent references with higher burstiness, while the purple nodes have higher centrality, which often appear at the junction of the two clusters and may represent a bridge between the two clustering topics. The highest node centrality links the themes “gastrointestinal microbial ecology” and “inflammatory bowel disease” [21] (Figure 5G and Appendix A). These studies compared the gastrointestinal microorganisms of IBD patients and a control group, and they found that IBD patients had abnormal gastrointestinal microflora and that correcting the microbial imbalance may contribute to remission of IBD.

Based on the references, the most emergent research in the past three years has introduced the global status of IBD and proposed the importance of the prevention and management of IBD [1]. One study proposed a dietary habit that can induce sustained remission in CD children and can produce changes in the fecal microbiota associated with remission [22], and another demonstrated that diet habits can induce IBD remission and that the microbiota is affected by dietary habits and is involved in the pathogenesis of IBD [23]. These findings are not only closely related to the topic of this study, but they also show that researchers are currently considering the mechanism of the influence of dietary habits on IBD from the perspective of the microbiome (Appendix A).

We found that most of the highly cited references, high-emergence references, and high-centrality references appeared after 2005 (Appendix A), indicating that the research results of great value in this field mostly appeared after this time, which is also consistent with the two stages of literature development discussed previously.

### 3.6. Keyword Analysis

#### 3.6.1. Keyword Frequency Analysis

The keywords represent the focus of a study. Analyzing keywords can identify the hotspots and trends of research in the field. The bibliometrix package was used to statistically analyze Keywords Plus and author keywords, obtaining 3133 Keywords Plus and 2683 author keywords. VOSviewer and CiteSpace were used to merge and analyze the two sets of keywords. VOSviewer analysis showed that there were 5334 keywords, and 446 keywords with a frequency greater than 5 were selected for visualization (Figure 6E). In CiteSpace, in order to include as much data as possible within the allowable range, the g-index (K = 34) was used to obtain a total of 987 keywords for subsequent analysis. In the keyword co-occurrence graph, the number of nodes is the number of keywords included in the analysis (Figure 6A). The node size represents the frequency of the keywords, and the connection between nodes represents two keywords appearing simultaneously in the same paper.

The frequency analysis of Keywords Plus, author keywords, and all keywords was performed and presented in the form of a word cloud (Figure 6B,C). Excluding the search terms, the top five keywords can be roughly divided into microorganisms, diet components, and risk descriptions. Among them, the number of occurrences of “microbiota” was the highest, and the frequencies of “chain fatty acid” and “risk” were also high. Considering the centrality, degree, and sigma (Σ) value of the nodes, after excluding the search terms, the most frequent occurrences were for “expression”, “NF-κB”, “chain fatty acid”, “dietary fiber”, and “celiac disease” associated with disease performance (Appendix A). In general, “microbiota” and “chain fatty acid” were at the forefront of multiple rankings and are important research topics in this field.

#### 3.6.2. Keyword Cluster Analysis

Clustering analysis of the keywords through software algorithms can help us further understand the topics in this research field. In CiteSpace, the default algorithm was used for clustering analysis, the LLR algorithm was used to mark the clustering, and the timeline diagram (Figure 6D) was drawn. The modularity Q value was 0.4424, indicating that the clustering network structure was good and the results were significant. The S value (weighted mean silhouette) was 0.7314, indicating that the clustering efficiency was high and the classification results are convincing. The clusters were divided into 12 modules (Figure 6D and Table 3), and 5 of them had more than 100 keywords. The largest cluster was labeled inflammatory bowel disease by the LLR algorithm, and it contained 151 keywords. The most frequent keywords were inflammatory bowel disease (576 times), Crohn’s disease (371 times), and ulcerative colitis (368 times). The second largest cluster had 143 keywords and was marked as gut microbiota. The most frequent keywords were gut microbiota (139 times), chain fatty acid (67 times), and intestinal microbiota (66). A total of 139 keywords in the third largest cluster were labeled as associated with Mediterranean-like dietary patterns, with the highest frequencies for risk (98), children (60), and epidemiology (54). The fourth and fifth clusters were labeled as nutritional therapy and inflammatory bowel disease, respectively. In general, these 12 clusters can be artificially divided into the following three categories: IBD and related diseases, such as inflammatory bowel disease, irritable bowel syndrome, canine chronic enteropathies, and celiac disease; dietary components and habits, such as Mediterranean-like dietary pattern association, nutritional therapy, and short-chain fatty acids; and pathogenesis, such as gut microbiota, intermediate biomarker, and virulence factor.

Then, the keywords were clustered in VOSviewer to verify the above categories. The six clusters (Figure 6E) can be summarized as four broad categories: inflammatory bowel disease and related diseases, eating habits and intestinal diseases, eating habits and disease risk management, and intestinal flora and intestinal diseases. This is basically consistent with the above CiteSpace clustering category.

We also used the bibliometrix package to perform factorial analysis on the Keywords Plus and author keywords. The number of clusters was set as four and the keyword clusters were drawn (Figure 6G,I) to further verify the clustering results. The closer the distance is between two points in the figure, the higher the frequency of occurrence in the same study, and the closer the keyword is to the center point, the more attention it receives in the field. The results of the Keywords Plus analysis showed that the keywords could be roughly divided into three categories and one subcategory (Figure 6G). The red word clusters were mainly related to IBD and intestinal microbes, among which gut microbiota, dietary patterns, and IBD were the closest nodes to the center point, indicating that they had received extensive attention in this field. Green word clusters were mainly related to healthy lifestyles, eating habits, diagnosis, and treatment. Blue word clusters were mainly related to eating habits and disease risk. Purple word clusters were mainly related to the study of pathogenesis. The author keyword analysis showed that there were two major categories and two subcategories (Figure 6I). The blue word cluster was mainly related to the risk factors and pathogenesis of IBD, the red word cluster was mainly related to the risk factors and related disease manifestations of IBD, and the other two word clusters were related to emotion and stress. In general, although the analysis results of VOSviewer and the bibliometrix packages were slightly different from those of CiteSpace, they were generally consistent.

#### 3.6.3. Keyword Evolution and Emergent Analysis

In order to obtain a preliminary understanding of the evolution of keywords, the OALM platform was used to analyze keywords year by year, and those with a frequency of occurrence were selected to draw the accumulation area map (Figure 6F). It can be seen that in the early stage before 2006, the research focused on eating habits and the disease itself, paying less attention to the mechanism. From 2007 onwards, the role of the gut microbiota gradually attracted attention. In recent years, dietary nutrition and intestinal ecological imbalance have also received more and more attention.

To further understand the evolution of the research direction, the bibliometrix package was used for thematic evolution analysis. The first stage of literature development before 2006 was selected as the first time slice, the last five years were selected as the third time slice, and the year between them was the second time slice. The topic classification of each time slice is visualized in Figure 6H. The size of each color block in the graph represents the number of keywords in the corresponding topic. The width of the connection between the two topics is proportional to the number of common keywords in the two topics and represents the degree of correlation between the two topics. The research topics showed a clear trend of dispersion to unity and then to dispersion in the three stages. In the first stage, the research topics focused on the disease itself, conventional treatments for IBD, dietary fiber, and the immune system. In the second stage, gut microbiota and disease risk factors became the focus of attention. In recent years, researchers have studied the effects of dietary habits on IBD from various aspects such as NF-κB, oxidative stress, bacteria, chain fatty acids, metabolism, and endoplasmic reticulum stress, which indicates that the research theory in this field is rapidly developing and constantly improving.

Using CiteSpace for emergent analysis, we can obtain more detailed research hotspots and frontier dynamics within a certain period of time and predict the research trend in the future. In Table 4, the year and outbreak period of each keyword can be seen. In the early stage of 1993–2023, the research hotspots in the field mainly focused on the clinical and pathological manifestations of IBD and related diseases. The research on eating habits and living habits was also relatively simple, focusing on topics such as fish oil, dietary fiber, aberrant crypt foci, and activity index. In the middle stage of IBD research, various pathophysiological processes related to the microbiota and IBD have gradually become the focus of research, including aspects such as the gut microbiome, NF-κB, innate immunity, regulatory T cells, and barrier function, which indicates that the research on the effect of dietary habits on IBD is deepening. In recent years, the research focus has shifted to the components of dietary patterns, and the pathogenesis has been studied at the subcellular and molecular levels. Meanwhile, research on the microbiota has also maintained a high degree of popularity. This is manifested in studies on carbohydrate diets, low-FODMAP diets, fecal microbiota transplantation, the gut–brain axis, cytokines, the Mediterranean diet, oxidative stress, polyunsaturated fatty acids, endoplasmic reticulum stress, and other keywords. It can be seen from the table that these research directions are still hot in 2023 and could be a short-term future research trend.

## 4. Discussion

The incidence and prevalence of inflammatory bowel disease have increased significantly throughout the world [24], which has created a huge medical and social burden. At present, the clinical treatment of inflammatory bowel disease focuses on the relief of symptoms after onset. The main therapeutic drugs are 5-aminosalicylic acid, immunosuppressive agents, corticosteroids, and biological agents that have emerged in recent years. However, many patients will have no response or no sustained response to these drugs, and even adverse reactions [25,26]. With the development of human civilization, scholars and the public have begun to seek disease prevention and treatment methods focused on the lifestyle. Dietary habits are an indispensable component of human life and are expected to become an important direction for the prevention and treatment of inflammatory bowel disease.

This article comprehensively analyzed the relevant literature on the impact of dietary patterns on IBD from the perspectives of the number of papers, authors, institutions, countries, journals, and references. The statistical results of the total number and annual number of papers were in line with the law of scientific literature growth. We also divided the development of this field into two stages: the slow development stage from 1993 to 2006, and the subsequent significant growth stage. It can be seen from the fitting curve that the number of related publications will grow rapidly in the foreseeable future. The number of articles published by the authors conformed to the generalized Lotka’s law. Among them, Ghosh S. and Gasbarrini A. had the highest academic comprehensive ranking and are excellent researchers in this field. Ghosh S. had a higher g-index. His representative work [27] expounds that the intestinal microbiome affects the human immune system, behavior, and mood and plays an important role in metabolic and inflammatory diseases such as IBD. Gasbarrini A.’s m-index and total citation number were more prominent. Although he started to carry out research in this field late in its development, he has already had a strong academic influence. One of his representative works introduces the differences in intestinal microflora among individuals, which are affected by factors such as diet, stress, disease, and antibiotic use. It can not only affect the immune system but can also lead to a variety of intestinal and extraintestinal diseases [28]. Another representative work reviews the effects of different food components and dietary habits on intestinal microorganisms [29], such as animal proteins, saturated fat, sugar, salt, additives, and other harmful intestinal microflora, which can also lead to changes in the intestinal barrier. Plant protein, ω-3, polyphenols, micronutrients, low-FODMAP, GFD, the Mediterranean diet, etc., were shown to be beneficial to intestinal health. In addition, it can be seen from the references that the research results of Lee D., Lewis J. D., and other researchers are widely recognized. The authors’ clustering analysis showed that the link between dietary patterns and IBD may be related to gut microbiota and lipids.

The United States is far ahead of other participating countries in terms of the number of articles, citations, and corresponding authors, while China has had strong momentum in article output in recent years and is the only Asian and developing country in the top five. Harvard University ranks first in the number of articles and node centrality among all institutions and is the most important institution in this research field. The clustering of 1706 institutions suggested that dietary habits may affect IBD by affecting intestinal permeability, intestinal flora balance, and the expression of inflammatory factor receptors. Natural dialectics showed that science and technology promote social development, and good social development is conducive to scientific and technological progress [30]. Based on the number of published articles and the intensity of mutual cooperation, the core countries and institutions were mostly from North America, Europe, and East Asia. These regions also have high levels of social and economic development, indicating that the number of studies, regional cooperation, and economic level have a significant relationship. This phenomenon is in line with the view of natural dialectics.

The literature included in this study has had a great influence on the field of global science and the field of interest [10]. The 15 most highly cited studies cover three main aspects: the global epidemiological status of IBD, studies on dietary habits that have a mitigating or aggravating effect on IBD, and studies on the relationship between dietary habits and the gut microbiota. Highly emergent and central references also roughly fell into these three categories. Among these core references, studies related to intestinal flora had the highest frequency, which indicates that intestinal microorganisms are an important direction for researchers to explore the mechanism of the effect of dietary patterns on IBD and may play an important mediating role between dietary patterns and the pathogenesis of IBD. From the results of the journal analysis, whether in the literature included in the analysis or in the references, the core journals had higher impact factors, most of which are JCR Q1 journals. This shows that this field’s research and knowledge sources tend to be published in high-impact journals. In addition to clinical medicine and basic medicine, the cited journals were also environmental, social, psychological, chemical, and other types of journals, suggesting that these factors may also play a role in the process of dietary patterns affecting IBD.

Visualizing the frequency and evolution of keywords in the literature included in the study, we found that the main keywords were “inflammatory bowel disease”, “Crohn’s disease”, “ulcerative colitis”, “diet”, “pattern”, “risk”, “inflammation”, “nutrition”, “gut microbiota”, “microbiota”, “microbiome”, “chain fatty acid”, “IBD”, “colitis”, “cancer”, “irritable bowel syndrome”, “probiotics”, “colorectal obesity”, “dysbiosis”, etc. According to the results of the keyword cluster analysis using multiple tools, the focus of the studies was to determine the dietary patterns that are beneficial or detrimental to IBD patients from the perspective of “nutritional therapy”, and to explore the specific mechanism of dietary patterns affecting IBD from the aspects of “gut microbiota”, “short-chain fatty acids” and “virulence factors”. Of course, as mentioned above, there were also some studies focusing on IBD-related diseases. These studies showed that the interaction between patients and their microbiome occurs in the early stage of the pathogenesis of CD [31]. In UC patients, “short-chain fatty acids” produced by intestinal microbial fermentation of dietary fiber interact with G protein-coupled receptor 43 (GPR43), which is beneficial to the treatment of UC and other inflammatory diseases [32]. Different dietary patterns will have beneficial or adverse effects on IBD patients, and the mechanism of the impact is closely related to intestinal microorganisms. High-sugar diets can lead to changes in intestinal microbial composition in mice, resulting in intestinal microflora dysfunction and barrier damage, promoting the development of IBD, through mechanisms such as increasing the abundance of Bacteroides fragilis and Akkermansia muciniphila, and eroding the colonic mucus layer. Highly saturated fatty acids can lead to a decrease in the tight junction proteins occludin and ZO-1, which impairs the integrity of the intestinal barrier [33]. Western dietary patterns are associated with an increased risk of IBD. Excessive consumption of fried foods, which are common in Western diets, such as fried chicken, can lead to a decrease in the UC protective flora Adlercreutzia and the flora that maintains intestinal homeostasis [34], while a “meat-eating” diet that involves the consumption of large quantities of poultry, red meat, and processed meat is associated with the development of UC [35]. Exclusive enteral nutrition (EEN) alone can induce mucosal healing and prolong remission in some children and adolescents with mild to moderate CD, but it is less effective in adult patients [36,37]. The emergent analysis showed that in the early stage of this research field, the focus was on the clinical and pathological features of IBD-related diseases such as “active Crohn’s disease”, “colon cancer”, and “aberrant crypt foci”, as well as basic dietary components such as “dietary fiber” and “fish oil”. In the middle stage, the research focused on the risk factors and pathophysiological processes of IBD. The gut microbiota gradually gained the attention of researchers, and this continues to be the case to this day. In recent years, studies have continuously explored the mechanism of dietary patterns’ effects on IBD at the subcellular and molecular levels and explored dietary patterns that are beneficial to IBD patients, specifically reflected in keywords such as “NF-κB”, “endoplasmic reticulum stress”, “oxidative stress”, and “Mediterranean diet”. NF-κB is an important regulator of the immune response. It has been found in many studies to be closely related to the occurrence and development of IBD [38,39] and can be affected by many factors such as drugs and intestinal microorganisms. For example, oxyberberine improves oxidative stress and mediates nuclear factor kappa B (NF-κB) pathway inhibition, significantly improving UC in rats [40]. The Mediterranean diet is a famous healthy diet, with a small amount of fat (mainly olive oil), candy, red meat, wine, moderate fish, poultry, eggs, and dairy products; it is rich in plant-based food as the main component and is characterized by high amounts of dietary fiber and low levels of saturated fatty acids [41]. Adhering to the Mediterranean diet reduces disease activity and related inflammatory biomarkers in IBD patients, improves the quality of life of patients, and also improves IBD-related diseases such as non-alcoholic fatty liver disease. This anti-inflammatory process is partially mediated by intestinal microorganisms [34,42].

In summary, it is a significant focus in this field to find dietary patterns that are beneficial to IBD patients, as well as the specific mechanisms by which various dietary patterns, such as the Mediterranean diet and Western diet, lead to changes in the intestinal microbiota, thereby affecting IBD. This is worthy of long-term attention from researchers as it provides a theoretical basis for the formation of more healthy eating patterns and living habits, thereby promoting the prevention and management of IBD, reducing the medical and social burden of IBD worldwide, and improving the quality of life of IBD patients.

A limitation of this study is that the literature retrieved from the WoSCC database may not be all the literature in this field, and the results of this analysis are relatively broad, which can only summarize the research hotspots and trends in the field and cannot specify the existing research results.

## 5. Conclusions

In this study, multiple analytical tools were used to perform bibliometric analysis on studies related to the effects of dietary habits on IBD, and the results were mutually verified to enhance the credibility. We summarized the development process and current trends of studies on the effects of dietary habits and IBD in the past 30 years. Reviewing all the analysis results, we found that a Mediterranean-like dietary pattern may be a dietary habit that is beneficial to IBD patients. Carbohydrates, fatty acids, and inulin-type fructans were dietary components closely related to IBD. Fatty acid, gut microbiota, NF-κB, oxidative stress, and endoplasmic reticulum stress were found to be hot spots in the current research field on the influence of dietary habits on IBD and are also the research trends of the future. These results can provide a theoretical basis for follow-up studies by researchers in this field.

## Figures and Tables

**Figure 1 nutrients-15-03442-f001:**
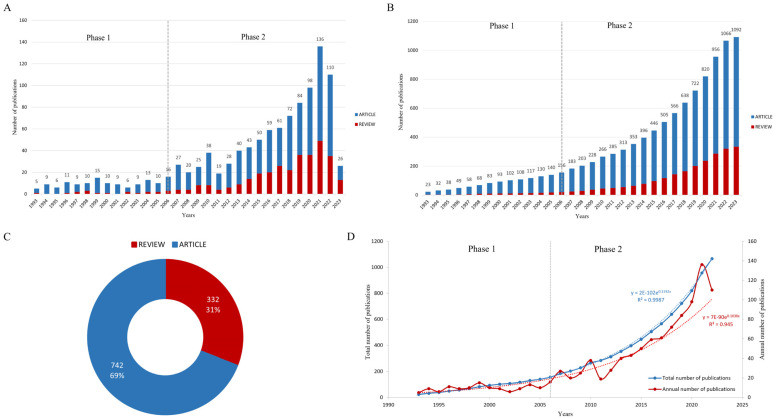
(**A**) Statistics on the number of publications per year; (**B**) statistics on the total number of publications; (**C**) statistics on the type of literature; (**D**) curve fitting analysis of Price’s literature growth curve.

**Figure 2 nutrients-15-03442-f002:**
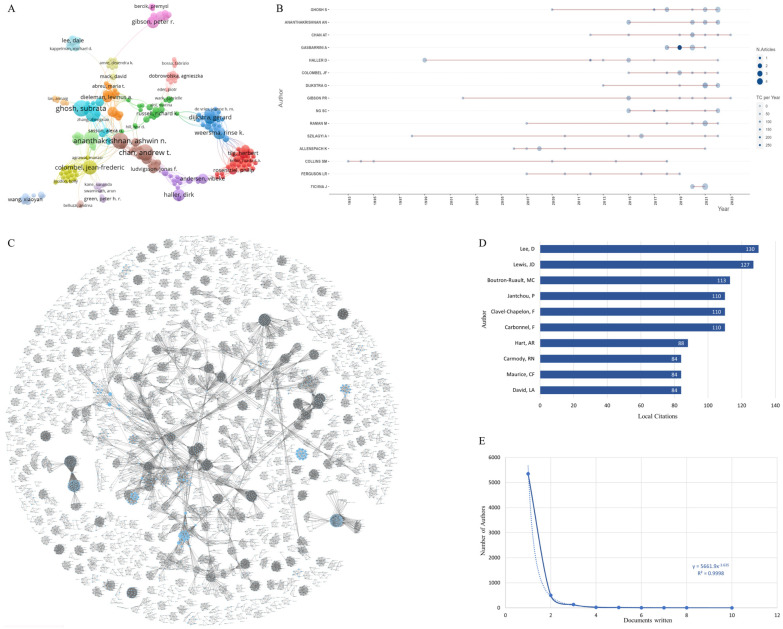
(**A**) Author co-occurrence network based on VOSviewer; (**B**) author’s production over time; (**C**) author collaboration network based on OALM; (**D**) top 10 citations among local authors; (**E**) Lotka’s law fitting curve.

**Figure 3 nutrients-15-03442-f003:**
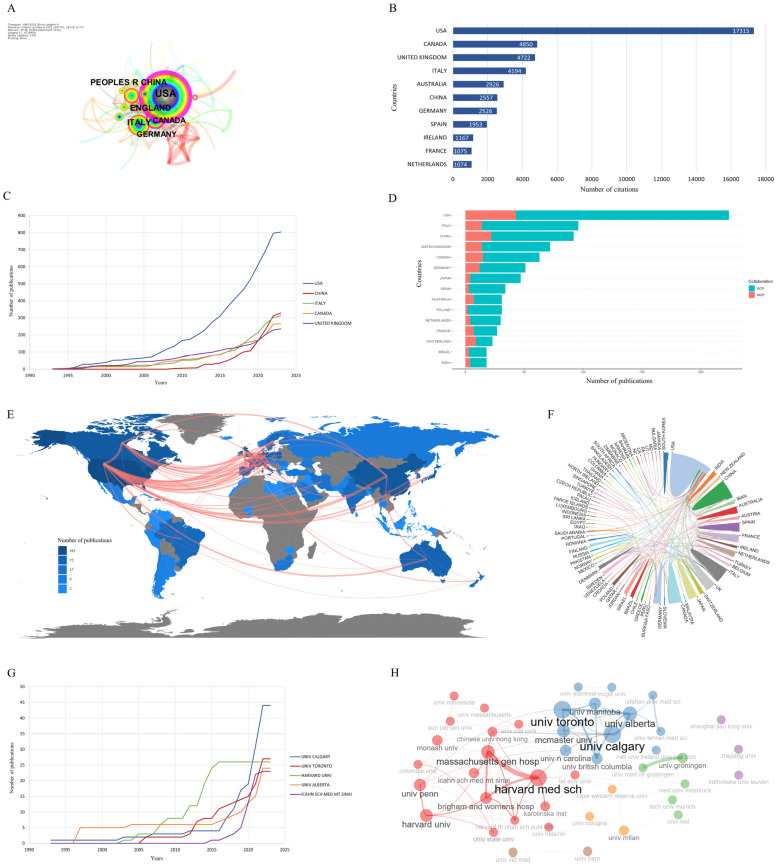
(**A**) the country co-occurrence network based on CiteSpace; (**B**) the top 10 countries cited; (**C**) the output of the top 5 countries participating in the number of articles changes over time; (**D**) the top 10 countries responsible for the number of studies (SCP: Single Country Publications, indicating that the authors of the article are from the same country; MCP: Multiple Country Publications, indicating that the author of the article is from multiple countries, that is, there is cooperation between countries); (**E**) geographical visualization of national output and cooperative relations (National color shade indicates the number of articles involved: the darker the color, the higher the number of articles involved, and the thicker the lines between countries, the closer the cooperation); (**F**) string graph of national output and cooperative relations (color block size indicates the number of articles issued: the larger the color block, the higher the number of articles involved, and the thicker the lines between the color blocks, the closer the cooperation between countries); (**G**) the output of the top 5 institutions participating in the number of articles changes over time; (**H**) the institutional cooperation network based on the bibliometrix package (each point represents an institution, and the thicker the inter-agency lines, the closer the cooperation).

**Figure 4 nutrients-15-03442-f004:**
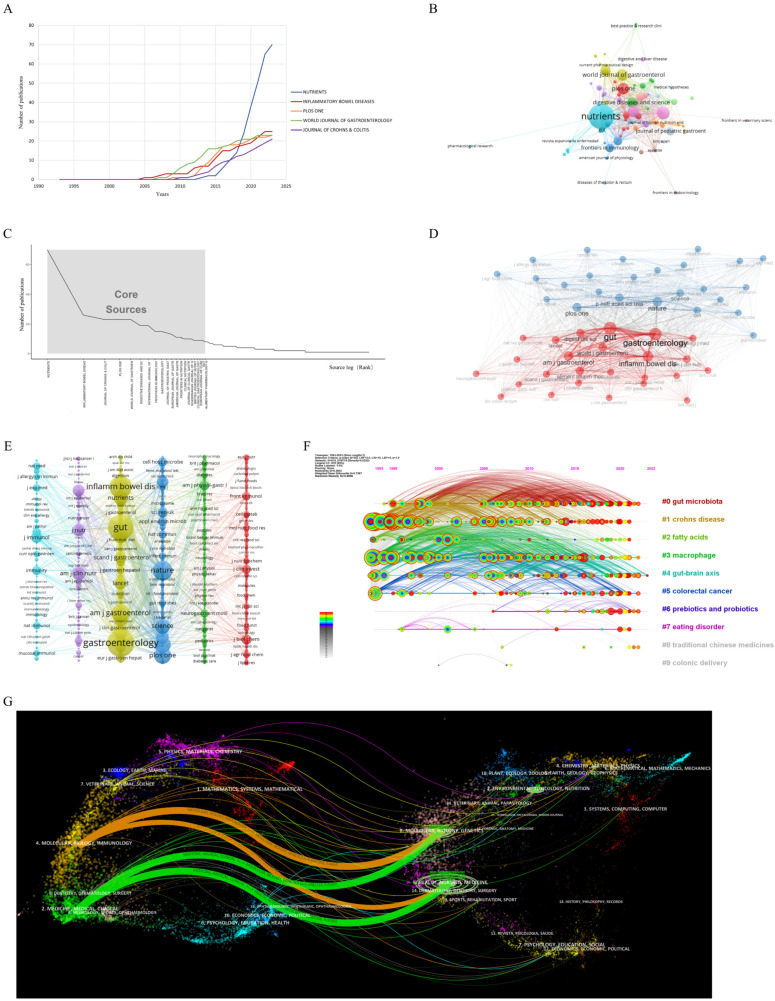
(**A**) The annual number of publications of the top five journals obtained based on the bibliometrix package; (**B**) the journal co-occurrence and cooperation network based on VOSviewer (each node represents a journal, the size of the node represents the number of publications, and the thickness of the lines between nodes represents the intensity of cooperation between journals); (**C**) Bradford’s law analysis based on the bibliometrix package; (**D**) co-occurrence and cooperative network analysis of cited journals based on bibliometrix package (each node represents a journal, the size of the journal represents the number of citations, and the thickness of the lines between nodes represents the co-citation intensity between journals); (**E**) co-occurrence and cooperative network analysis of cited journals based on VOSviewer package (each node represents a journal, the size of the journal represents the number of citations, and the thickness of the lines between nodes represents the co-citation intensity between journals); (**F**) clustering timeline diagram of cited journals based on CiteSpace (each node represents a journal, and each horizontal line represents a cluster); (**G**) dual-map overlay analysis based on CiteSpace (different colors on the left side represent different types of cited journals, and different colors on the right side represent different types of cited journals).

**Figure 5 nutrients-15-03442-f005:**
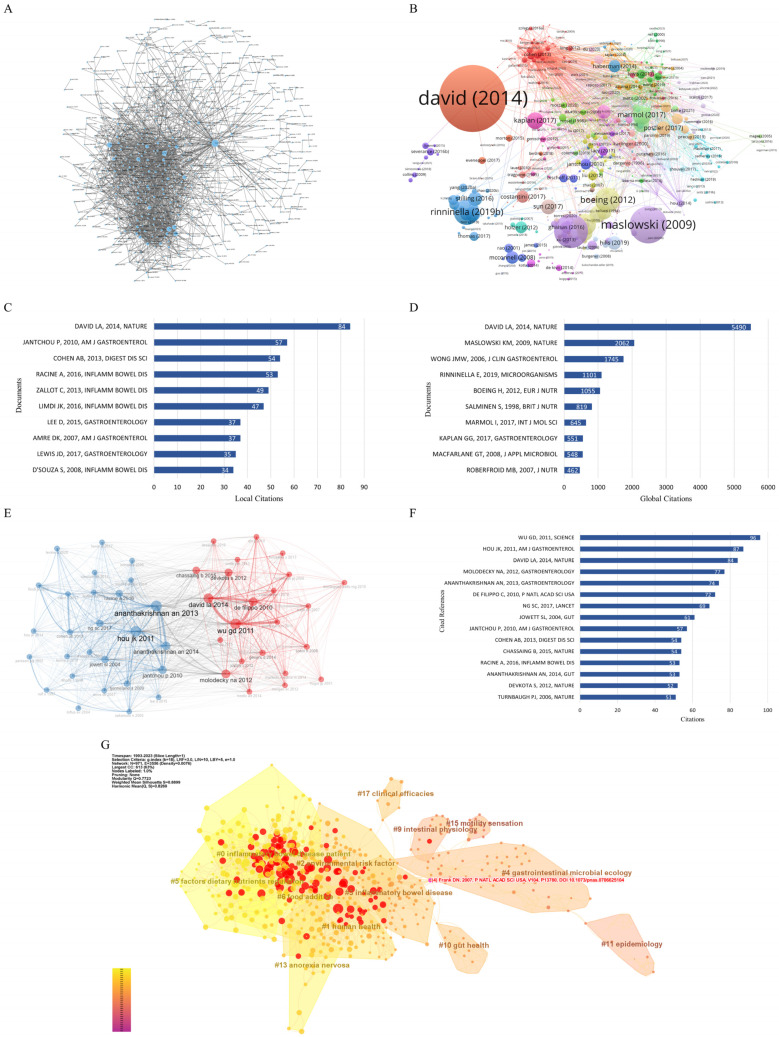
(**A**) Literature co-occurrence and cooperation network based on OALM; (**B**) literature co-occurrence and cooperation network based on VOSviewer (each node represents a piece of literature, and the connection between nodes represents inter-literature cooperation); (**C**) top 10 local citations in the 1074 articles included in the analysis; (**D**) top 10 global citations in the 1074 articles included in the analysis; (**E**) reference co-occurrence and cooperation network based on bibliometrix (each node represents a reference, the node size represents the number of references cited, and the connection between nodes represents the cooperation between references); (**F**) top 15 references cited; (**G**) reference clustering based on CiteSpace (red nodes represent references with high burstiness, and purple nodes represent references with high centrality).

**Figure 6 nutrients-15-03442-f006:**
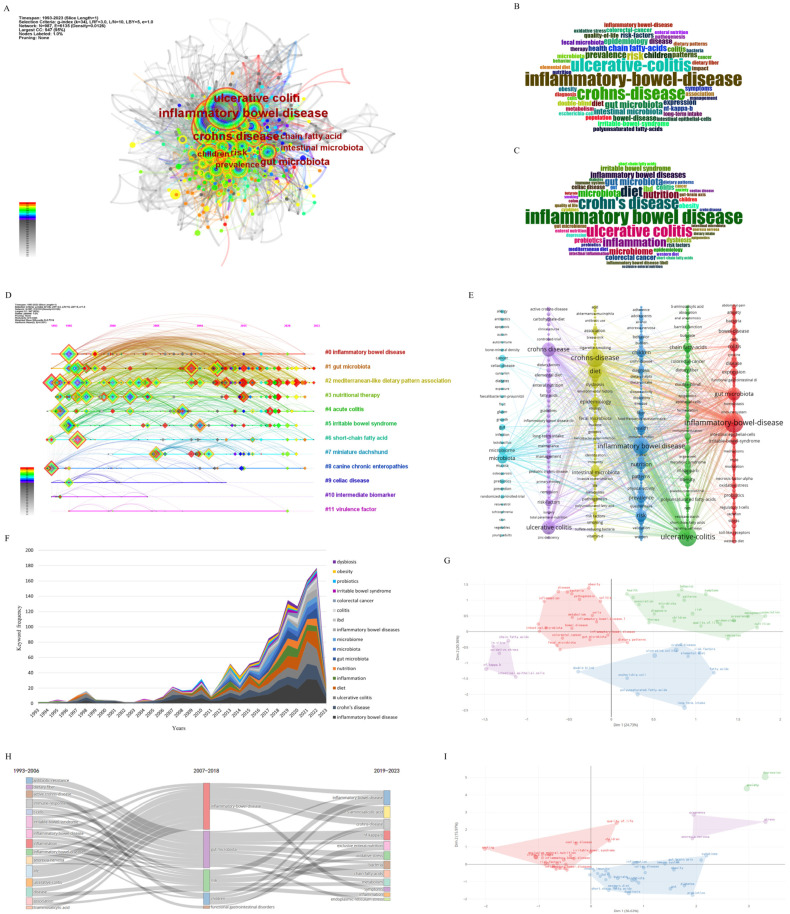
(**A**) Keyword co-occurrence network based on CiteSpace (each node represents a keyword, the node size represents the frequency of keywords, and the connection between nodes represents the co-occurrence of keywords); (**B**) Keywords Plus word cloud based on bibliometrix package; (**C**) author keywords word cloud based on bibliometrix package; (**D**) keyword clustering timeline map based on CiteSpace (each node represents a keyword, each red node represents a keyword with high burstiness, and each horizontal line represents keyword clustering); (**E**) keyword co-occurrence network map based on VOSviewer (each node represents a keyword, different color vertical lines represent keyword clustering, and connections between nodes represent keyword co-occurrence); (**F**) annual frequency analysis of important keywords based on OALM; (**G**) factorial analysis of Keywords Plus based on the bibliometrix package (each color block represents a keyword classification); (**H**) thematic evolution analysis of Keywords Plus based on the bibliometrix package; (**I**) factorial analysis of author keywords based on the bibliometrix package (each color block represents a keyword classification).

**Table 1 nutrients-15-03442-t001:** Top 10 authors under multiple influence indicators based on bibliometrix package.

Rank	Author	h_Index	Author	g_Index	Author	m_Index	Author	Total Citation
1	ANANTHAKRISHNAN AN	7	GHOSH S	10	GASBARRINI A	1.167	GASBARRINI A	1546
2	COLOMBEL JF	7	ANANTHAKRISHNAN AN	8	DAY AS	1	HALLER D	1182
3	GASBARRINI A	7	GASBARRINI A	8	PETERS V	1	NG SC	793
4	GHOSH S	7	HALLER D	8	ADOLPH TE	1	LEWIS JD	684
5	HALLER D	7	CHAN AT	8	ANANTHAKRISHNAN AN	0.778	GHOSH S	675
6	CHAN AT	6	COLOMBEL JF	7	COLOMBEL JF	0.778	LEE D	575
7	DIJKSTRA G	6	DIJKSTRA G	7	KIM J	0.75	COLLINS SM	374
8	FERGUSON LR	6	NG SC	7	NG SC	0.667	COLOMBEL JF	334
9	KIM J	6	RAMAN M	7	DIJKSTRA G	0.545	ANANTHAKRISHNAN AN	313
10	NG SC	6	SZILAGYI A	7	CHAN AT	0.5	TILG H	286

**Table 2 nutrients-15-03442-t002:** Twenty-one core journals based on Bradford’s law.

Journals	Rank	Number of Publications	Percentage of All Publications	IF	JCR Partition
*Nutrients*	1	70	6.52%	6.706	1
*Inflammatory Bowel Diseases*	2	26	2.42%	7.290	1
*Journal of Crohns & Colitis*	3	23	2.14%	10.02	1
*PLoS ONE*	4	23	2.14%	3.752	2
*World Journal of Gastroenterology*	5	23	2.14%	5.374	2
*Digestive Diseases and Sciences*	6	19	1.77%	3.487	3
*International Journal of Molecular Sciences*	7	19	1.77%	6.208	1
*Frontiers in Immunology*	8	15	1.40%	8.786	1
*Gastroenterology*	9	15	1.40%	33.883	1
*Journal of Pediatric Gastroenterology and Nutrition*	10	14	1.30%	3.288	2
*European Journal of Gastroenterology & Hepatology*	11	13	1.21%	2.586	4
*American Journal of Gastroenterology*	12	11	1.02%	12.045	1
*Frontiers in Nutrition*	13	11	1.02%	6.59	1
*Clinical Nutrition*	14	10	0.93%	7.643	1
*Journal of Clinical Gastroenterology*	15	10	0.93%	3.147	4
*Scandinavian Journal of Gastroenterology*	16	10	0.93%	3.027	4
*British Journal of Nutrition*	17	9	0.84%	4.125	3
*Clinical Gastroenterology and Hepatology*	18	9	0.84%	13.576	1
*European Journal of Clinical Nutrition*	19	9	0.84%	4.884	2
*Gut*	20	9	0.84%	31.793	1
*Alimentary Pharmacology & Therapeutics*	21	8	0.74%	9.524	1

**Table 3 nutrients-15-03442-t003:** Clusters based on CiteSpace.

ClusterID	Size	Silhouette	Label (LLR)	Average Year	Top Keywords (Frequency)
0	151	0.639	inflammatory bowel disease (633.98, 1.0 × 10^−4^)	2005	inflammatory bowel disease (576) Crohn’s disease (371) ulcerative colitis (368)
1	143	0.658	gut microbiota (1443.69, 1.0 × 10^−4^)	2012	gut microbiota (139) chain fatty acid (67) intestinal microbiota (66)
2	134	0.64	Mediterranean-like dietary pattern association (360.41, 1.0 × 10^−4^)	2013	risk (98) children (60) epidemiology (54)
3	104	0.755	nutritional therapy (462.4, 1.0 × 10^−4^)	2005	pattern (54) diet (50) colitis (45)
4	104	0.739	inflammatory bowel disease (671.64, 1.0 × 10^−4^)	2007	expression (43) Nf κ b (38) oxidative stress (20)
5	72	0.737	irritable bowel syndrome (1404.33, 1.0 × 10^−4^)	2008	prevalence (66) irritable bowel syndrome (58) quality of life (38)
6	70	0.818	short-chain fatty acid (374.44, 1.0 × 10^−4^)	2008	colorectal cancer (58) colon cancer (23) aberrant crypt foci (9)
7	64	0.849	miniature dachshund (461.16, 1.0 × 10^−4^)	2010	intestinal epithelial cell (26) intestinal inflammation (21) dendritic cell (15)
8	50	0.881	canine chronic enteropathies (221.53, 1.0 × 10^−4^)	2002	disease (45) diagnosis (18) acid (13)
9	27	0.942	celiac disease (162.71, 1.0 × 10^−4^)	2001	celiac disease (31) autoantibody (3) antigen (3)
10	17	0.945	intermediate biomarker (64.9, 1.0 × 10^−4^)	1995	supplementation (7) cell proliferation (4) cytokine production (3)
11	11	0.969	virulence factor (52.83, 1.0 × 10^−4^)	2004	anal anastomosis (6) bowel (5) ileal pouch (2)

**Table 4 nutrients-15-03442-t004:** Emergent analysis of keywords based on CiteSpace (Light blue: keywords do not appear; dark blue: keywords appear; red: keywords are clearly emerging).

Keywords	Year	Strength	Begin	End	1993–2023
colon cancer	1993	3.19	1993	2012	▃▃▃▃▃▃▃▃▃▃▃▃▃▃▃▃▃▃▃▃ ▂▂▂▂▂▂▂▂▂▂▂
epithelial cell proliferation	1993	3.17	1993	1998	▃▃▃▃▃▃ ▂▂▂▂▂▂▂▂▂▂▂▂▂▂▂▂▂▂▂▂▂▂▂▂▂
fish oil	1994	5.18	1994	2001	▂ ▃▃▃▃▃▃▃▃ ▂▂▂▂▂▂▂▂▂▂▂▂▂▂▂▂▂▂▂▂▂▂
absorption	1994	3.56	1994	2008	▂ ▃▃▃▃▃▃▃▃▃▃▃▃▃▃▃ ▂▂▂▂▂▂▂▂▂▂▂▂▂▂▂
cell proliferation	1994	2.16	1994	2007	▂ ▃▃▃▃▃▃▃▃▃▃▃▃▃▃ ▂▂▂▂▂▂▂▂▂▂▂▂▂▂▂▂
dietary fiber	1995	4.53	1995	2010	▂▂ ▃▃▃▃▃▃▃▃▃▃▃▃▃▃▃▃ ▂▂▂▂▂▂▂▂▂▂▂▂▂
colonic mucosa	1995	3.68	1995	2013	▂▂ ▃▃▃▃▃▃▃▃▃▃▃▃▃▃▃▃▃▃▃ ▂▂▂▂▂▂▂▂▂▂
elemental diet	1995	2.43	1995	2003	▂▂ ▃▃▃▃▃▃▃▃▃ ▂▂▂▂▂▂▂▂▂▂▂▂▂▂▂▂▂▂▂▂
controlled trial	1996	3.63	1996	2007	▂▂▂ ▃▃▃▃▃▃▃▃▃▃▃▃ ▂▂▂▂▂▂▂▂▂▂▂▂▂▂▂▂
epithelial cell	1996	3.07	1996	2013	▂▂▂ ▃▃▃▃▃▃▃▃▃▃▃▃▃▃▃▃▃▃ ▂▂▂▂▂▂▂▂▂▂
active Crohn’s disease	1996	2.61	1996	2005	▂▂▂ ▃▃▃▃▃▃▃▃▃▃ ▂▂▂▂▂▂▂▂▂▂▂▂▂▂▂▂▂▂
distal ulcerative coliti	1996	2.44	1996	2003	▂▂▂ ▃▃▃▃▃▃▃▃ ▂▂▂▂▂▂▂▂▂▂▂▂▂▂▂▂▂▂▂▂
carcinogenesis	1996	2.17	1996	2008	▂▂▂ ▃▃▃▃▃▃▃▃▃▃▃▃▃ ▂▂▂▂▂▂▂▂▂▂▂▂▂▂▂
cigarette smoking	1998	3.81	1998	2010	▂▂▂▂▂ ▃▃▃▃▃▃▃▃▃▃▃▃▃ ▂▂▂▂▂▂▂▂▂▂▂▂▂
smoking	1998	3.02	1998	2007	▂▂▂▂▂ ▃▃▃▃▃▃▃▃▃▃ ▂▂▂▂▂▂▂▂▂▂▂▂▂▂▂▂
expression	1994	2.75	1999	2008	▂ ▂▂▂▂▂ ▃▃▃▃▃▃▃▃▃▃ ▂▂▂▂▂▂▂▂▂▂▂▂▂▂▂
dietary factor	2000	2.73	2000	2010	▂▂▂▂▂▂▂ ▃▃▃▃▃▃▃▃▃▃▃ ▂▂▂▂▂▂▂▂▂▂▂▂▂
bone mineral density	2001	2.15	2001	2016	▂▂▂▂▂▂▂▂ ▃▃▃▃▃▃▃▃▃▃▃▃▃▃▃▃ ▂▂▂▂▂▂▂
aberrant crypt foci	2002	2.79	2002	2014	▂▂▂▂▂▂▂▂▂ ▃▃▃▃▃▃▃▃▃▃▃▃▃ ▂▂▂▂▂▂▂▂▂
restorative proctocolectomy	2004	2.44	2004	2009	▂▂▂▂▂▂▂▂▂▂▂ ▃▃▃▃▃▃ ▂▂▂▂▂▂▂▂▂▂▂▂▂▂
supplementation	1994	2.41	2004	2007	▂ ▂▂▂▂▂▂▂▂▂▂ ▃▃▃▃ ▂▂▂▂▂▂▂▂▂▂▂▂▂▂▂▂
remission	2004	2.26	2004	2010	▂▂▂▂▂▂▂▂▂▂▂ ▃▃▃▃▃▃▃ ▂▂▂▂▂▂▂▂▂▂▂▂▂
activity index	2004	2.18	2004	2014	▂▂▂▂▂▂▂▂▂▂▂ ▃▃▃▃▃▃▃▃▃▃▃ ▂▂▂▂▂▂▂▂▂
irritable bowel syndrome	1995	3	2006	2007	▂▂ ▂▂▂▂▂▂▂▂▂▂▂ ▃▃ ▂▂▂▂▂▂▂▂▂▂▂▂▂▂▂▂
mast cell	2006	2.61	2006	2014	▂▂▂▂▂▂▂▂▂▂▂▂▂ ▃▃▃▃▃▃▃▃▃ ▂▂▂▂▂▂▂▂▂
Nf κ b	2008	5.4	2008	2010	▂▂▂▂▂▂▂▂▂▂▂▂▂▂▂ ▃▃▃ ▂▂▂▂▂▂▂▂▂▂▂▂▂
active ulcerative colitis	2008	2.48	2008	2018	▂▂▂▂▂▂▂▂▂▂▂▂▂▂▂ ▃▃▃▃▃▃▃▃▃▃▃ ▂▂▂▂▂
intestinal permeability	2008	2.09	2008	2016	▂▂▂▂▂▂▂▂▂▂▂▂▂▂▂ ▃▃▃▃▃▃▃▃▃ ▂▂▂▂▂▂▂
innate immunity	2009	4.95	2009	2016	▂▂▂▂▂▂▂▂▂▂▂▂▂▂▂▂ ▃▃▃▃▃▃▃▃ ▂▂▂▂▂▂▂
intestinal epithelial cell	2005	3.36	2009	2016	▂▂▂▂▂▂▂▂▂▂▂▂ ▂▂▂▂ ▃▃▃▃▃▃▃▃ ▂▂▂▂▂▂▂
gene expression	2010	4.2	2010	2016	▂▂▂▂▂▂▂▂▂▂▂▂▂▂▂▂▂ ▃▃▃▃▃▃▃ ▂▂▂▂▂▂▂
dendritic cell	2005	3.62	2010	2015	▂▂▂▂▂▂▂▂▂▂▂▂ ▂▂▂▂▂ ▃▃▃▃▃▃ ▂▂▂▂▂▂▂▂
celiac disease	1995	3.1	2011	2014	▂▂ ▂▂▂▂▂▂▂▂▂▂▂▂▂▂▂▂ ▃▃▃▃ ▂▂▂▂▂▂▂▂▂
immune response	1997	2.95	2011	2014	▂▂▂▂ ▂▂▂▂▂▂▂▂▂▂▂▂▂▂ ▃▃▃▃ ▂▂▂▂▂▂▂▂▂
diet-induced obesity	2012	3.61	2012	2017	▂▂▂▂▂▂▂▂▂▂▂▂▂▂▂▂▂▂▂ ▃▃▃▃▃▃ ▂▂▂▂▂▂
escherichia coli	1997	3.52	2013	2018	▂▂▂▂ ▂▂▂▂▂▂▂▂▂▂▂▂▂▂▂▂ ▃▃▃▃▃▃ ▂▂▂▂▂
adipose tissue	2013	3.36	2013	2017	▂▂▂▂▂▂▂▂▂▂▂▂▂▂▂▂▂▂▂▂ ▃▃▃▃▃ ▂▂▂▂▂▂
rheumatoid arthritis	2013	3.21	2013	2016	▂▂▂▂▂▂▂▂▂▂▂▂▂▂▂▂▂▂▂▂ ▃▃▃▃ ▂▂▂▂▂▂▂
anxiety-like behavior	2013	2.85	2013	2017	▂▂▂▂▂▂▂▂▂▂▂▂▂▂▂▂▂▂▂▂ ▃▃▃▃▃ ▂▂▂▂▂▂
necrosis factor-alpha	2013	2.49	2013	2019	▂▂▂▂▂▂▂▂▂▂▂▂▂▂▂▂▂▂▂▂ ▃▃▃▃▃▃▃ ▂▂▂▂
t-cell	2013	2.41	2013	2017	▂▂▂▂▂▂▂▂▂▂▂▂▂▂▂▂▂▂▂▂ ▃▃▃▃▃ ▂▂▂▂▂▂
food allergy	2014	3.17	2014	2014	▂▂▂▂▂▂▂▂▂▂▂▂▂▂▂▂▂▂▂▂▂ ▃ ▂▂▂▂▂▂▂▂▂
aryl hydrocarbon receptor	2014	3.15	2014	2017	▂▂▂▂▂▂▂▂▂▂▂▂▂▂▂▂▂▂▂▂▂ ▃▃▃▃ ▂▂▂▂▂▂
regulatory t-cell	2014	2.38	2014	2014	▂▂▂▂▂▂▂▂▂▂▂▂▂▂▂▂▂▂▂▂▂ ▃ ▂▂▂▂▂▂▂▂▂
visceral hypersensitivity	2014	2.25	2014	2016	▂▂▂▂▂▂▂▂▂▂▂▂▂▂▂▂▂▂▂▂▂ ▃▃▃ ▂▂▂▂▂▂▂
diversity	2015	4.94	2015	2019	▂▂▂▂▂▂▂▂▂▂▂▂▂▂▂▂▂▂▂▂▂▂ ▃▃▃▃▃ ▂▂▂▂
exclusive enteral nutrition	2015	3.81	2015	2019	▂▂▂▂▂▂▂▂▂▂▂▂▂▂▂▂▂▂▂▂▂▂ ▃▃▃▃▃ ▂▂▂▂
gastrointestinal symptom	2015	3.41	2015	2019	▂▂▂▂▂▂▂▂▂▂▂▂▂▂▂▂▂▂▂▂▂▂ ▃▃▃▃▃ ▂▂▂▂
maintenance therapy	2015	3.03	2015	2019	▂▂▂▂▂▂▂▂▂▂▂▂▂▂▂▂▂▂▂▂▂▂ ▃▃▃▃▃ ▂▂▂▂
prospective cohort	2015	2.39	2015	2016	▂▂▂▂▂▂▂▂▂▂▂▂▂▂▂▂▂▂▂▂▂▂ ▃▃ ▂▂▂▂▂▂▂
intestinal microbiota	2009	6.52	2016	2019	▂▂▂▂▂▂▂▂▂▂▂▂▂▂▂▂ ▂▂▂▂▂▂▂ ▃▃▃▃ ▂▂▂▂
fecal microbiota	2012	5.45	2016	2018	▂▂▂▂▂▂▂▂▂▂▂▂▂▂▂▂▂▂▂ ▂▂▂▂ ▃▃▃ ▂▂▂▂▂
colorectal cancer	1996	5.02	2016	2018	▂▂▂ ▂▂▂▂▂▂▂▂▂▂▂▂▂▂▂▂▂▂▂▂ ▃▃▃ ▂▂▂▂▂
metabolic syndrome	2016	3.7	2016	2019	▂▂▂▂▂▂▂▂▂▂▂▂▂▂▂▂▂▂▂▂▂▂▂ ▃▃▃▃ ▂▂▂▂
dysbiosis	2017	3.49	2017	2018	▂▂▂▂▂▂▂▂▂▂▂▂▂▂▂▂▂▂▂▂▂▂▂▂ ▃▃ ▂▂▂▂▂
faecalibacterium prausnitzii	2017	3.33	2017	2019	▂▂▂▂▂▂▂▂▂▂▂▂▂▂▂▂▂▂▂▂▂▂▂▂ ▃▃▃ ▂▂▂▂
gut microbiome	2014	2.81	2017	2018	▂▂▂▂▂▂▂▂▂▂▂▂▂▂▂▂▂▂▂▂▂ ▂▂▂ ▃▃ ▂▂▂▂▂
barrier function	2008	2.74	2017	2018	▂▂▂▂▂▂▂▂▂▂▂▂▂▂▂ ▂▂▂▂▂▂▂▂▂ ▃▃ ▂▂▂▂▂
environmental risk factor	2018	2.62	2018	2020	▂▂▂▂▂▂▂▂▂▂▂▂▂▂▂▂▂▂▂▂▂▂▂▂▂ ▃▃▃ ▂▂▂
gastrointestinal disorder	2018	2.62	2018	2020	▂▂▂▂▂▂▂▂▂▂▂▂▂▂▂▂▂▂▂▂▂▂▂▂▂ ▃▃▃ ▂▂▂
environmental factor	2012	2.38	2018	2020	▂▂▂▂▂▂▂▂▂▂▂▂▂▂▂▂▂▂▂ ▂▂▂▂▂▂ ▃▃▃ ▂▂▂
autism spectrum disorder	2018	2.12	2018	2019	▂▂▂▂▂▂▂▂▂▂▂▂▂▂▂▂▂▂▂▂▂▂▂▂▂ ▃▃ ▂▂▂▂
carbohydrate diet	2019	4.04	2019	2023	▂▂▂▂▂▂▂▂▂▂▂▂▂▂▂▂▂▂▂▂▂▂▂▂▂▂ ▃▃▃▃▃
low-FODMAP diet	2019	3.93	2019	2023	▂▂▂▂▂▂▂▂▂▂▂▂▂▂▂▂▂▂▂▂▂▂▂▂▂▂ ▃▃▃▃▃
diagnosis	1993	3.59	2019	2020	▂▂▂▂▂▂▂▂▂▂▂▂▂▂▂▂▂▂▂▂▂▂▂▂▂▂ ▃▃ ▂▂▂
physical activity	2019	3.4	2019	2023	▂▂▂▂▂▂▂▂▂▂▂▂▂▂▂▂▂▂▂▂▂▂▂▂▂▂ ▃▃▃▃▃
quality of life	2010	3.33	2019	2020	▂▂▂▂▂▂▂▂▂▂▂▂▂▂▂▂▂ ▂▂▂▂▂▂▂▂▂ ▃▃ ▂▂▂
fecal microbiota transplantation	2019	2.99	2019	2020	▂▂▂▂▂▂▂▂▂▂▂▂▂▂▂▂▂▂▂▂▂▂▂▂▂▂ ▃▃ ▂▂▂
Stress	2010	2.71	2019	2021	▂▂▂▂▂▂▂▂▂▂▂▂▂▂▂▂▂ ▂▂▂▂▂▂▂▂▂ ▃▃▃ ▂▂
enteral nutrition	1996	2.55	2019	2019	▂▂▂ ▂▂▂▂▂▂▂▂▂▂▂▂▂▂▂▂▂▂▂▂▂▂▂ ▃ ▂▂▂▂
pathogenesis	2015	2.45	2019	2023	▂▂▂▂▂▂▂▂▂▂▂▂▂▂▂▂▂▂▂▂▂▂ ▂▂▂▂ ▃▃▃▃▃
metaanalysis	2014	2.36	2019	2019	▂▂▂▂▂▂▂▂▂▂▂▂▂▂▂▂▂▂▂▂▂ ▂▂▂▂▂ ▃ ▂▂▂▂
gut–brain axis	2019	2.23	2019	2021	▂▂▂▂▂▂▂▂▂▂▂▂▂▂▂▂▂▂▂▂▂▂▂▂▂▂ ▃▃▃ ▂▂
validity	2019	2.22	2019	2019	▂▂▂▂▂▂▂▂▂▂▂▂▂▂▂▂▂▂▂▂▂▂▂▂▂▂ ▃ ▂▂▂▂
management	2013	2.14	2019	2020	▂▂▂▂▂▂▂▂▂▂▂▂▂▂▂▂▂▂▂▂ ▂▂▂▂▂▂ ▃▃ ▂▂▂
nutritional status	1995	2.11	2019	2020	▂▂ ▂▂▂▂▂▂▂▂▂▂▂▂▂▂▂▂▂▂▂▂▂▂▂▂ ▃▃ ▂▂▂
dietary pattern	2004	4.85	2020	2023	▂▂▂▂▂▂▂▂▂▂▂ ▂▂▂▂▂▂▂▂▂▂▂▂▂▂▂▂ ▃▃▃▃
fiber	2018	4.64	2020	2023	▂▂▂▂▂▂▂▂▂▂▂▂▂▂▂▂▂▂▂▂▂▂▂▂▂ ▂▂ ▃▃▃▃
protein	2020	3.19	2020	2023	▂▂▂▂▂▂▂▂▂▂▂▂▂▂▂▂▂▂▂▂▂▂▂▂▂▂▂ ▃▃▃▃
microbiota	2013	3.01	2020	2023	▂▂▂▂▂▂▂▂▂▂▂▂▂▂▂▂▂▂▂▂ ▂▂▂▂▂▂▂ ▃▃▃▃
cohort	2000	2.8	2020	2021	▂▂▂▂▂▂▂ ▂▂▂▂▂▂▂▂▂▂▂▂▂▂▂▂▂▂▂▂ ▃▃ ▂▂
cytokine	2000	2.56	2020	2021	▂▂▂▂▂▂▂ ▂▂▂▂▂▂▂▂▂▂▂▂▂▂▂▂▂▂▂▂ ▃▃ ▂▂
burden	2020	2.48	2020	2023	▂▂▂▂▂▂▂▂▂▂▂▂▂▂▂▂▂▂▂▂▂▂▂▂▂▂▂ ▃▃▃▃
gluten-free diet	1999	2.2	2020	2023	▂▂▂▂▂▂ ▂▂▂▂▂▂▂▂▂▂▂▂▂▂▂▂▂▂▂▂▂ ▃▃▃▃
questionnaire	2018	2.15	2020	2023	▂▂▂▂▂▂▂▂▂▂▂▂▂▂▂▂▂▂▂▂▂▂▂▂▂ ▂▂ ▃▃▃▃
Mediterranean diet	2019	7.91	2021	2023	▂▂▂▂▂▂▂▂▂▂▂▂▂▂▂▂▂▂▂▂▂▂▂▂▂▂ ▂▂ ▃▃▃
oxidative stress	2018	5.03	2021	2023	▂▂▂▂▂▂▂▂▂▂▂▂▂▂▂▂▂▂▂▂▂▂▂▂▂ ▂▂▂ ▃▃▃
risk	1993	4.85	2021	2023	▂▂▂▂▂▂▂▂▂▂▂▂▂▂▂▂▂▂▂▂▂▂▂▂▂▂▂▂ ▃▃▃
prevalence	1995	4.65	2021	2023	▂▂ ▂▂▂▂▂▂▂▂▂▂▂▂▂▂▂▂▂▂▂▂▂▂▂▂▂▂ ▃▃▃
marker	2021	3.92	2021	2021	▂▂▂▂▂▂▂▂▂▂▂▂▂▂▂▂▂▂▂▂▂▂▂▂▂▂▂▂ ▃ ▂▂
vitamin d	2015	3.17	2021	2023	▂▂▂▂▂▂▂▂▂▂▂▂▂▂▂▂▂▂▂▂▂▂ ▂▂▂▂▂▂ ▃▃▃
fecal calprotectin	2021	2.69	2021	2023	▂▂▂▂▂▂▂▂▂▂▂▂▂▂▂▂▂▂▂▂▂▂▂▂▂▂▂▂ ▃▃▃
depression	2004	2.39	2021	2021	▂▂▂▂▂▂▂▂▂▂▂ ▂▂▂▂▂▂▂▂▂▂▂▂▂▂▂▂▂ ▃ ▂▂
cardiovascular disease	2013	2.16	2021	2023	▂▂▂▂▂▂▂▂▂▂▂▂▂▂▂▂▂▂▂▂ ▂▂▂▂▂▂▂▂ ▃▃▃
gut microbiota	2012	6.39	2022	2023	▂▂▂▂▂▂▂▂▂▂▂▂▂▂▂▂▂▂▂ ▂▂▂▂▂▂▂▂▂▂ ▃▃
polyphenol	2022	2.79	2022	2023	▂▂▂▂▂▂▂▂▂▂▂▂▂▂▂▂▂▂▂▂▂▂▂▂▂▂▂▂▂ ▃▃
polyunsaturated fatty acid	2002	2.63	2022	2023	▂▂▂▂▂▂▂▂▂ ▂▂▂▂▂▂▂▂▂▂▂▂▂▂▂▂▂▂▂▂ ▃▃
fatty acid	1996	2.61	2022	2023	▂▂▂ ▂▂▂▂▂▂▂▂▂▂▂▂▂▂▂▂▂▂▂▂▂▂▂▂▂▂ ▃▃
long-term intake	2015	2.48	2022	2023	▂▂▂▂▂▂▂▂▂▂▂▂▂▂▂▂▂▂▂▂▂▂ ▂▂▂▂▂▂▂ ▃▃
endoplasmic reticulum stress	2019	2.11	2022	2023	▂▂▂▂▂▂▂▂▂▂▂▂▂▂▂▂▂▂▂▂▂▂▂▂▂▂ ▂▂▂ ▃▃

## Data Availability

All data generated or analyzed during this study are included in this published article and were derived from the following resources, available in the public domain: the Web of Science Core Collection of Clarivate Analytics (https://clarivate.com/, accessed on 18 May 2023). The bibliometrix package is used to test that the data quality and the data quality is reliable.

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
