# Peer review of "Effect of Dietary Patterns on Inflammatory Bowel Disease: A Machine Learning Bibliometric and Visualization Analysis"

_nutrients, 2023, doi:10.3390/nu15153442_

Round 1
Reviewer 1 Report
This paper used machine learning to examine world trends in articles on the relationship between inflammatory bowel disease (IBD) and dietary pattern. This paper is very useful in understanding the current relationship between IBD and diet and is well analyzed in a systematic manner. This study is very interesting, but has some problems.
Major comments;
#1. I find the paper difficult to understand in the first place. In addition, almost all of the figures were small in size, and it was difficult to distinguish some of the text. Please make them more legible.
Minor comments;
#1. In P6 L161, the number of authors is followed by the term "the number of authors", is that a correct expression?
#2. The authors wrote “The country with the highest average citation number of articles is AUSTRALIA” in L198. However, I could not find any evidence from the figure that Australia is the highest average citation.
Author Response
Response to Reviewer 1 Comments
First of all, we thank the reviewer 1 for these positive and constructive comments and suggestions.
Point 1: I find the paper difficult to understand in the first place. In addition, almost all of the figures were small in size, and it was difficult to distinguish some of the text. Please make them more legible.
Response 1: We are so sorry for the confusing display. First of all, for the difficulty in understanding this paper, it may be due to the lack of logic in our expression and some low-level grammatical errors, because our native language is not English, for which we are deeply sorry. Now we have revised the paper to make its logic more reasonable and layout more appropriate. In addition, we have used the language editing service recommended by the journal (order number: english- 69137) to allow researchers whose native language is English to revise our paper, so there should be no grammatical problems at present. In addition, as for the problem that all the figures are small and difficult to distinguish the text, we originally had high definition versions of all the figures, but after the figures are inserted into the word document, their clarity will automatically become blurred, thus causing difficulties for reviewers to read. Now we have uploaded high definition versions of all the figures into the submission system as a compressed package, please check. Thank you very much for your comments.
Point 2: In P6 L161, the number of authors is followed by the term "the number of authors", is that a correct expression?
Response 2: It is highly appreciated that the reviewer has raised this issue, and we are so sorry for the confusing expression. We recognize that this was a duplicate spelling error, which we have now corrected accordingly and used the journal's recommended English editing service (order number: english- 69137), so there should now be no relevant errors.
Point 3: The authors wrote “The country with the highest average citation number of articles is AUSTRALIA” in L198. However, I could not find any evidence from the figure that Australia is the highest average citation.
Response 3: It is highly appreciated that the reviewer has put forward this imperative issue. We are deeply sorry for this problem. It is our negligence. At present, we have added the corresponding table (Supplement Table 2) to quantify that Australia is the country with the highest average citation.
In all, I found the reviewer’s comments are quite helpful, and I revised my paper point-by-point. Thank you and the review again for your help!
Best regards,
Yours Sincerely

Reviewer 2 Report
In this study, the authors analyzed the international literature as far as the effect of dietary patterns on IBD from the perspective of the number of publications, authors, institutions, countries, journals, and bibliographic references concerned. Multiple analytical tools were used to perform bibliometric analysis such as the bibliometrix package and VOSviewer. The authors summarized the current trends in studies on the effects of dietary habits and IBD over the past 30 years.
It is an interesting and useful study that used innovative tools and reached significant conclusions regarding the present and future of research in the field of the effect of dietary habits on IBD.
The work can be published taking into account the correction of some typographical oversights (e.g. lines 53, 332, 591, etc.).
It is also suggested that authors could list separately the abbreviations used in the study.
See comments to the authors
Author Response
Response to Reviewer 2 Comments
First of all, we thank the reviewer 2 for these positive and constructive comments and suggestions.
Point 1: The work can be published taking into account the correction of some typographical oversights (e.g. lines 53, 332, 591, etc.).
Response 1: We are so sorry for the confusing display. We have now corrected accordingly and used the journal's recommended English editing service (order number: english- 69137), so there should now be no relevant errors.
Point 2: It is also suggested that authors could list separately the abbreviations used in the study.
Response 2: We appreciate the reviewer’s attention to this important comment. We have added an "abbreviations" section to the manuscript, where all the abbreviations involved in the manuscript are listed and explained.
In all, I found the reviewer’s comments are quite helpful, and I revised my paper point-by-point. Thank you and the review again for your help!
Best regards,
Yours Sincerely
